# Profile Autonomous Underwater Vehicle System for Offshore Surveys

**DOI:** 10.3390/s23073722

**Published:** 2023-04-04

**Authors:** Bin Jiang, Zhenhua Xu, Shuguo Yang, Yonghua Chen, Qiang Ren

**Affiliations:** 1College of Electromechanical and Engineering, Qingdao University of Science and Technology, Qingdao 266061, China; jiangbin@qdio.ac.cn; 2Key Laboratory of Ocean Circulation and Waves, Institute of Oceanology, Chinese Academy of Sciences, Qingdao 266071, China; 3College of Mathematics and Physics, Qingdao University of Science and Technology, Qingdao 266061, China

**Keywords:** marine vehicles, profile measurement, sea profile, underwater vehicles

## Abstract

Offshore marine engineering, offshore aquaculture, and offshore environmental protection require periodic offshore surveys. At present, the main means of offshore marine surveys are mooring buoys and marine survey ships. Anchored buoys are fixed in place for a long time, which affects the navigation of ships. Therefore, mooring buoys cannot be deployed over a large area with high density. The cost of marine survey ships is high, especially when multipoint synchronous marine surveys are needed, and marine survey ships cannot perform offshore surveys under bad sea conditions. A profile autonomous underwater vehicle system is developed to meet the requirements of multipoint short-term synchronous offshore surveys. It is a small, reusable, low-cost equipment designed to move up and down at a mooring position while measuring temperature, salinity, depth, and other quantities along a vertical water section. Profile autonomous underwater vehicles can be commanded remotely and report their measurements in near real-time via wireless telemetry. The time it takes for a profile AUV to move up and down can indicate the current velocity. Tests were carried out on a wharf and in offshore areas, and the results were satisfactory.

## 1. Introduction

Offshore marine areas account for 8% of the total ocean area and 0.5% of the total volume, but they provide 90% of the marine fishing biomass [1]. Almost all sea oil and gas exploitation come from offshore marine environments [2,3]. Offshore surveys are an indispensable tool for disaster prevention and reduction, ensuring maritime traffic safety, marine engineering, coastal engineering construction, marine resource development, marine economic development, national security, and so on [4,5,6]. Marine autonomous systems also have an increasing range of applications in the defense, industry, and policy sectors, such as geohazard assessment associated with oil and gas infrastructure [7,8,9].

Sea surface investigations can be completed using satellites or buoys [10,11]. Marine profile surveys are most often completed using submerged buoys [12]. The spacing of the measuring equipment on submerged buoys is relatively sparse, making it difficult to meet offshore survey profile resolution requirements [13]. At present, most offshore profile surveys rely on survey ships [14]. Historically, the density in the space and time of oceanographic observations has been limited by the cost of operating ships [15].

Oceanographers have been largely limited by cost to relatively few platforms from which to examine the offshore environment [16]. Autonomous underwater vehicles (AUVs) have positioned themselves as essential for maritime exploration due to their operational efficiency, autonomy, and relatively low operational cost [17,18,19].

At present, the main purpose of AUV research is to focus on the needs of deep-sea resource exploration and submarine target search [20,21]. The Remote Environmental Monitoring UnitS (REMUS) series of underwater vehicles, designed and produced by the Woods Hole Institute of Oceanography in the United States, are shown in Figure 1 [22,23]. Among them, the REMUS 6000 successfully located the engine wreckage of Air France 447 in 2010 [24]. The REMUS 100 is a convenient underwater vehicle with a length of 1.6 m, a diameter of 0.19 m, and a weight of 36.5 kg [25]. In addition to the conventional configuration of the communication module, energy module, control module, and drive module, the REMUS 100 series can also carry optional devices, which can be adjusted in real-time according to the mission needs [26]. The REMUS 100 emerged in the early stage of the Gulf War and participated in the United States combat mission in Iraq [25]. China’s sailfish underwater vehicle, shown in Figure 2, has basically completed the initial design objectives and met the expected functional requirements [27].

At present, AUV research mainly focuses on route planning and control in the horizontal direction. In offshore surveys, the application of AUVs is still relatively small. Offshore surveys usually require high-density horizontal and vertical synchronous data. According to the characteristics of offshore surveys, this research designs an autonomous underwater vehicle suitable for offshore profile measurement. It is fixed at a point, using an anchor system, to complete the vertical profile measurement. It is commanded remotely and reports its measurements in near real-time via wireless telemetry. Due to the low cost, multiple systems can be deployed horizontally over a wide range.

## 2. System Description

During an offshore survey, the current velocity may be significant, and a profile AUV needs to maintain a fixed-point profile measurement in this current. Therefore, combined with the characteristics that a submerged buoy can maintain a fixed point in the current velocity and an AUV can move freely, a set of profile AUV systems that can complete a profile measurement at fixed points and move along a cable are designed. By analyzing the hydrodynamic environment, the shape, power, and control of the profile AUVs are designed. We carried out an underwater data return experiment on a wharf and carried out an experiment with a system running in a strong current in an offshore survey. The experiments achieved satisfactory results.

### 2.1. Overall Design

As shown in Figure 3, the profile AUV system consists of a floating ball (1), upper damping block (2), Kevlar cable (3), profile AUV (4), lower damping block (5), tensioning hammer (6), and fixed anchors (7). The floating ball and tensioning hammer tighten the Kevlar cable into a straight line to provide a track for the profile movement of the AUV. The upper damping block and lower damping block limit the operation range of the profile AUV. The fixed anchor fixes the whole system at the observation point. The profile AUV is designed to be in a microgravity state and connects with the Kevlar cable through two sliding connectors. The profile AUV can slide up and down along the Kevlar cable through the sliding connectors. When the profile AUV is not operating, it stops at the lower damping block along the Kevlar cable. When the profile AUV (4) operates, the conduit propeller starts. Under the thrust of the propeller, the profile AUV moves upward along the Kevlar cable. After reaching the specified depth, or where there is no change in the depth of the profile AUV within a period of time, the propeller stops working. Under the action of microgravity, the profile AUV moves along the Kevlar cable to the lower damping block. During this period, marine measuring equipment is used to collect marine parameters. When it reaches the lower damping block, the measuring equipment stops operating.

### 2.2. Profile AUV Shape Design

Underwater self-propelled vehicles navigate underwater and carry their own energy for operational tasks. Due to the limited internal space and low energy-carrying capacity, improving navigation efficiency and reducing energy consumption are of great significance to underwater self-propelled vehicles. At present, streamlined revolving bodies are widely used in the design of underwater self-propelled vehicles because of their advantages, such as simple structure, easy processing, and low navigation resistance. Optimizing the shape of underwater vehicles and reducing resistance is an effective means to improve their endurance. At present, the main body of AUVs has an axisymmetric body shape, and the design method uses a curve. On the basis of using an appropriate linear external shape parameter model, the flow field is calculated to obtain the required hydrodynamic parameters, and the optimal shape of a certain hydrodynamic parameter is obtained through a design optimization method. Common shapes include the Myring, water drop, and spindle. The Myring shape has been widely used because of its low resistance, low energy consumption, large plot ratio, and low manufacturing cost.

As shown in Figure 4, the Myring forward flow line, a, reduces resistance; b is the middle, equipped with energy, control, etc.; and c is the stern that maintains the overall balance and provides power. The Myring shape is widely used. For example, the famous REMUS and Maya underwater vehicles use this form. The main reason is that the Myring shape has excellent hydrodynamic performance and has been verified through practical engineering. Second, the axisymmetric body structure is easy to process and build. The parallel middle section can adopt a modular structure, and the module settings can be adjusted according to different tasks.

In order to obtain small profile AUV resistance, the following principles and requirements should be followed in the selection of the overall configuration: small resistance and good hydrodynamic performance, facilitation of the layout of each module system, and good processing performance. In the preliminary design stage, the general layout is mainly divided into three parts. The bow provides buoyancy and reduces forward resistance. In the middle, according to different mission requirements, appropriate mission equipment and its supporting power supply module are selected, the propeller control system is arranged, and batteries required by various systems and equipment inside the underwater vehicle are arranged. The stern is equipped with a ducted propeller.

The profile AUV designed in this study combines the advantages of the existing popular AUV shape design. The bow adopts a spherical shape, similar to the Myring bow, and the middle adopts a parallel section. It does not need a tail because the profile AUV rotates around a Kevlar cable. The structure of the profile AUV is shown in Figure 5. The bow (1) uses a floating ball to provide buoyancy for the whole AUV. The spherical shape is similar to the streamlined shape. The resistance in water is small, and the spherical shape requires little additional processing. The floating ball passes through a fixed column (2) in the middle, which is easily fixed. The fixed column provides a connection for the whole AUV. The bow, middle cabin (3), and stern duct propellers (4) are fixed on the fixed column to form a rigid connection and provide support for the power transmission of the AUV. The middle cabin includes an internal battery, controller, propeller driver, and temperature-depth sensor (5), which is the core part of the profile AUV. The stern tube propeller is the power part of the profile AUV. It is controlled by the middle cabin to provide power for the profile AUV. The temperature-depth sensor is fixed on the middle cabin to provide the temperature and depth parameters required for basic profile measurement, as well as the control parameters for the controller. There is one sliding connector (6) in the middle and one at the stern, as shown in Figure 6. The Kevlar cable passes through the middle of the two sliding connectors to ensure that the profile AUV is fixed on the Kevlar cable and that the profile AUV can slide along the Kevlar cable easily. A fixed connector (7) is used to fix the standby profile measuring equipment, which can be either self-contained or directly read. The profile AUV provides three interfaces and can be connected to three pieces of direct reading equipment.

The connector between the profile AUV and Kevlar cable is shown in Figure 7. The connector is divided into two layers, each of which is a parallel casing. The inner and outer walls of the casing are smooth and can slide directly with the fixed shaft. When the Kevlar cable is in contact with the sleeve, the white plastic column can rotate freely. It produces very little friction, which minimizes obstruction to the up-and-down movement of the profile AUV. The cable contacts the AUV in any direction, which makes the AUV move up and down with less friction resistance.

### 2.3. Profile AUV Power Analysis

There is a positive correlation between the underwater resistance and the speed of an underwater vehicle. Overcoming the water resistance and maintaining the speed depend on the power provided by the propulsion device. The factors affecting the underwater vehicle speed mainly come from two aspects: one is the resistance, and the second is the thruster thrust. The thrust is related to the thruster motor power and propulsion efficiency, while the resistance is related to the overall underwater vehicle dimensions. Excessive increases in the thruster motor power and propulsion efficiency will affect the overall underwater vehicle dimensions. Therefore, on the one hand, speed optimization should consider optimizing the shape and reducing the resistance; on the other hand, it should consider propeller power matching.

According to the propeller propulsion form, underwater vehicles can be divided into the water jet type, horizontal rotation type, channel type, magnetic fluid type, tandem type, Kurt conduit type, and conduit propeller type. According to different functional requirements and environmental characteristics, different forms of propellers can be selected to meet the requirements of scheduled tasks. Among propulsion methods, the underwater propulsion efficiency of a ducted propeller is the highest.

A single propeller structure design is adopted, which is the most popular design, but its propulsion efficiency is low, and the cavitation phenomenon is obvious. A more advanced ducted propeller can be used for propulsion. A ducted propeller is also called a casing propeller. As opposed to an ordinary propeller, there is a sleeve outside. The section of the ducted propeller is divided into a broken line and a wing. Compared with an ordinary propeller, a ducted propeller can improve the Wakefield at the stern and increase the thrust and propulsion efficiency.

In order to improve the hydrodynamic and cavitation performance of the propeller, a sleeve is usually used to wrap the propeller, acting as its guide tube. The ducted propeller adds a ducted structure outside the propeller to separate the propeller flow field to form an internal and external basin. The incoming flow is pre-accelerated to reduce the capacity loss of the propeller tail vortex and the slight loss of the blade to improve the propulsion efficiency. After the fluid passes through the duct, the flow velocity increases and the flow velocity reaching the propeller blade surface is increased, which makes it easier for the propeller to spiral accelerate the fluid.

The stress of the profile AUV in the vertical direction is shown in Figure 8. Since the designed profile AUV is in a microgravity state, it can be considered that the gravity and buoyancy offset are not marked in the figure. In the working state, the profile AUV is subjected to a vertical upward thrust *F* from the propeller, the friction *F* on the Kevlar cable, and the water resistance in the vertical direction. Among them, the calculation of the water resistance *F* on the profile AUV during movement can be deduced using the Morison equation:(1)$$F_{y}=\frac{1}{2}\rho C_{dy}A_{y}v_{y}{}^{2}$$

The water resistance on the profile AUV in the vertical direction is directly proportional to the square of the water density, resistance coefficient, projected area, and velocity. The drag coefficient is related to the shape of an object. Here, a simple cylindrical drag coefficient of 0.85 is selected. Since the bow uses a ball with a diameter of 20 cm, the projected area is 0.0314 m^2^, and the density of water is 1 × 10^3^ kg/m^3^. Considering the sampling frequency of offshore survey profile data, the speed *v_y_* is set to 1 m/s. Therefore, the maximum water resistance of profile AUV in the vertical direction is:(2)$$F_{y}=0.5\times 10^{3}\times 0.85\times 0.0314\times 1^{2}=13.345\;N$$

The profile AUV is mainly affected by the force of the water flow in the horizontal direction, which can also be deduced using the Morison equation:(3)$$F_{x}=\frac{1}{2}\rho C_{dx}A_{x}v_{x}{}^{2}$$

In the horizontal direction, the force on the profile AUV is directly proportional to the water force and the water density *ρ*, the resistance coefficient *C_dx_*, the projected area *A_x_*, and the square of the velocity *v_x_*, as shown in Figure 9. The resistance coefficient *C_dx_* is related to the shape of the object. Here, a simple cylindrical resistance coefficient of 0.85 is selected. Because the bow uses a ball with a diameter of 20 cm, the length of the profile AUV is 1 m, and the projection of the AUV on the horizontal force is approximately rectangular, so the projection area *A_x_* is 0.2 m^2^, the density of water is 1 × 10^3^ kg/m^3^, and the velocity *v_x_* is the velocity of the current. Based on experience, the offshore current velocity is generally less than 1 m/s, and the velocity *v_x_* is set to 1 m/s. Therefore, the maximum force that the profile AUV is subjected to due to water flow in the horizontal direction is:(4)$$F_{x}=0.5\times 10^{3}\times 0.85\times 0.2\times 1^{2}=85\;N.$$

Due to the force between the tension hammer and the floating ball, it can be considered that the Kevlar cable is always in a tight vertical state, and the horizontal force between the Kevlar cable and the profile AUV is converted to a vertical friction force. There is sliding friction between the profile AUV connector and the Kevlar cable. Since the sliding friction coefficient is usually less than 0.1, the sliding friction coefficient is set to 0.1, and the maximum friction force along the cable system is:(5)$$f=\mu \times F_{y}=0.1\times 85=8.5\;N.$$

The forces acting on the profile AUV in the vertical direction include the water resistance *F_y_* and the friction force *f* on the Kevlar cable. The water resistance *F_y_* is the dynamic force varying with the vertical speed *v_y_* of the profile AUV, and the friction force *f* on the Kevlar cable is related to the speed of the water flow. Since the water-flow-velocity change process is not abrupt, the friction force f on the Kevlar cable over a short time can be regarded as a constant force. Therefore, the power of the profile AUV can be designed to be slightly greater than the water resistance *F_y_*. When the current velocity is 0, the profile AUV will advance at an ideal speed of 1 m/s. When the current velocity is 1 m/s, the power of the profile AUV is greater than the friction force f on the Kevlar cable, so the profile AUV will advance at a low speed, which can also meet the profile data acquisition frequency of an offshore survey. Here, the propeller thrust is 14 N, and the corresponding propeller power is 40 W. Due to the use of a ducted propeller, the thrust is generally increased by approximately 20% [27]; that is, the power F of the profile AUV can be maintained at 18.2 N to ensure that the profile AUV can move at a speed of 0~1 m/s when the current velocity is lower than 1 m/s. Since the current velocity on the ocean surface is usually larger than that at the bottom of the ocean, to ensure that the profile AUV can initially move smoothly, the profile AUV is considered to be in a microgravity state in the water, and it stops at the limit close to the seabed when there is no power.

### 2.4. Profile AUV Control Design

The control of the profile AUV is different from that of an ordinary AUV. It only needs to complete a one-dimensional movement in the profile direction. This simple control can ensure the reliability of the profile AUV. The profile AUV needs to collect multisensory data at the same time. Therefore, the Linux operating system is installed on the profile AUV controller to complete the concurrent tasks (Figure 10).

The profile AUV is equipped with temperature and depth sensors to provide reference data for power control. When the profile AUV reaches the predetermined near-sea depth, it stops its propeller. Under microgravity, the profile AUV will slowly sink along the Kevlar cable to the limit near the seabed. When the profile AUV is started, and the depth does not change, it will control the propeller to reverse and stop the profile AUV from moving upward as the Kevlar cable is entangled with water, grass, fishing net, etc.

A depth sensor is required to collect depth information during operations continuously. At the same time, *n* sensors continuously collect data to obtain profile data. The signal strength of the wireless communication module is checked to determine whether the data can be sent. When the signal strength reaches the sending strength, the analyzed and sorted data are sent to the receiving system of the shore station. The data sent each time are the data collected during the last free fall, including depth information, information collected by each sensor, and flow rate.

In order to ensure continuous depth acquisition using the depth sensor, data can be collected by multiple sensors and sent by wireless communication at the same time. It is necessary to utilize an operating system on the control equipment to ensure continuous depth data acquisition. The data can be collected by multiple sensors and sent by wireless communication at the same time. Linux with high stability is adopted for use. The depth sensor continuous depth acquisition task, multiple sensor acquisition data task, and wireless communication transmission data task are treated as separate processes, and the tasks are independent and concurrent with each other.

As shown in Figure 11, the relationship between the propeller power and thrust is similar to a quadratic function relationship, and it is more energy-saving to operate with a small power level for a long time than with a large power level for a short time.

For example, with 50 W power over 4 s, the propeller thrust work is
(6)$$F*t=3\times 4=12\;\left(\mathrm{lbf}\cdot s\right).$$

With 200 W power over 1 s, the propeller thrust work is
(7)$$F*t=8\times 1=8\;\left(\mathrm{lbf}\cdot s\right).$$

It can be seen from the above that with the same power consumption, more propeller work can be obtained using less power. However, the power cannot be infinitely small. It is necessary to ensure that the propeller thrust is greater than the microgravity. Therefore, it is necessary to use a PID algorithm to control the propeller speed to ensure that the AUV rises at a low speed with a stable state, which can save energy and complete more profile observations. The conventional PID controller is frequently employed in AUV applications, including commercial and scientific uses, due to its simple and effective structure [28,29,30].
(8)$$u\left(k\right)=K_{P}\cdot e\left(k\right)+K_{I}\cdot \sum_{i=0}e\left(i\right)+K_{D}\cdot \left[e\left(k\right)-e\left(k-1\right)\right]$$
Proportion: $e\left(k\right)$ error for this time

Integral: $\sum e\left(i\right)$ accumulation of errors

Differential: $e\left(k\right)-e\left(k-1\right)$ the difference between this error and the last error

Because there are error integrals $\sum e\left(i\right)$ that are accumulated all the time, the current output $u\left(k\right)$ is related to all past states (Figure 12).

As shown in Figure 13, NORMAL means that the propeller works at the same speed all the time. It takes 20 s for the AUV to rise, the speed keeps increasing, and the maximum speed is 0.75 m/s. PID means that the AUV is controlled by a PID algorithm. It sets the speed to 0.4 m/s and controls the speed of the AUV to rise steadily at 0.4 m/s, which takes 24 s. When it does not reach 0.4 m/s, the speed of the AUV continues increasing, and when it reaches 0.4 m/s, the AUV will rise steadily at 0.4 m/s. In Figure 13b, the vertically upward direction is positive, and the vertically downward speed is negative. The faster the AUV moves in the water, the greater the resistance it experiences and the more energy it will consume. To verify the energy-saving effect of the PID control, the AUV will work under NORMAL and PID until the power is insufficient after being fully charged. NORMAL works 103 times, and PID works 134 times, so PID saves approximately 30% more energy than NORMAL.

## 3. Experiments

To verify the feasibility of the system, experiments were carried out at the wharf of the West Coast Park of the Chinese Academy of Sciences. The remote data transmission and remote command communication functions were verified in the wharf experiment. To verify the practicability and reliability of the system, an offshore experiment was completed in an offshore survey.

### 3.1. Wharf Experiment

Figure 14a shows the West Coast Park wharf of the Institute of Oceanography, Chinese Academy of Sciences. The profile AUV system is lowered above the culvert of the wharf (Figure 14b). To facilitate recycling, the Kevlar cable is tied to the wharf above the culvert. The tension hammer is attached to the other end of the cable. The tension hammer is suspended above the seabed. There is a limit near the tension hammer. When the profile AUV is not operating, it stops at this limit. When the profile AUV is operating, it moves along the tensioned cable. The profile AUV is adjusted to the microgravity state. The profile AUV moves upward and downward along the Kevlar cable during operation.

Figure 15 shows the fall process in the wharf experiment. The results of the profile AUV depth measurement can reflect the tide. The water depth of the wharf is approximately 6 m at the lowest tide and approximately 10 m at the highest tide, which is consistent with the actual measurement. During high tide and low tide, the current velocity is very small, and the lines are relatively parallel. With increasing water depth, the time required for the profile AUV to fall increases correspondingly. In the middle part of the tide, near a water depth of 7.5 m, the curves intersect. This is because even though the depth is smaller, the fall time becomes longer due to the large flow velocity and the increased cable resistance.

When the profile AUV rises to the water surface, it transmits the data back to the shore station receiving system in near real-time through wireless telemetry, and the shore station receiving system sends commands to the profile AUV. Figure 16 shows the maximum water depth of the profile AUV test every hour after deployment. When deployment, the acquisition frequency is set once an hour. After 3 h of deployment, the acquisition frequency is remotely set to twice an hour through remote communication, corresponding to the box part in Figure 16. After 6 h of deployment, the acquisition frequency is remotely set to once an hour through remote communication. In the wharf experiment, the remote data transmission and remote-command communication functions have been well-verified.

### 3.2. Offshore Survey

The offshore survey area is selected near Weihai, China (Figure 17). It is necessary to complete the measurement of the profile temperature and flow velocity for a tidal cycle. To carry out comparative experiments, the profile AUV (Figure 18b) and an acoustic Doppler current profiler (ADCP) (Figure 18a) are fixed on a survey ship, and a CTD is manually lowered to the seabed every hour and then recycled. The water depth at this position is approximately 12 m. The ADCP and the profile AUV operate twice every hour. The profile AUV stops working 1 m away from the water surface.

As shown in Figure 19, different from the wharf experiment, the deepest depth of the profile AUV does not change with the tide fluctuation because the top of the wharf experimental Kevlar cable is fixed on the wharf, and the top of the offshore survey experimental Kevlar cable is fixed on the survey ship, which changes with the tide fluctuation.

On 20 April 2021, and 21 April 2021, a comparative profile AUV and manual lowering CTD temperature-depth map experiment was carried out. Here, time points at 3–4 h intervals were selected for temperature-depth comparisons. Because manual lowering is dangerous at night, no manual measurement was carried out at night. The temperature and depth changes of the profile AUV and manually lowered CTD are relatively consistent (Figure 20), and the profile AUV measurement time is significantly longer than the manually lowered CTD measurement (Figure 21). At 15:00 on 20 April 2021, the time difference between the two measurements was the smallest, and the temperature-depth variation curve was also relatively consistent. At 8:00 on 21 April 2021, the time difference between the two measurements was the largest. At this time, the current velocity was very large, and the CTD manual lowering was significantly accelerated, resulting in a change in the measuring instrument too late, leading to obvious measurement errors.

## 4. Discussion

Because the profile AUV is used in the offshore survey, the offshore water depth is generally shallow, and the current velocity at full depth is basically the same (Figure 22). Therefore, it is considered that the cable friction force on the profile AUV is constant during the descent process.

The profile AUV falling process force analysis (Figure 23) shows that the cable friction force $f^{\prime }$ on the profile AUV is stable during each falling process, and the microgravity ${mg}^{\prime }$ is also stable. The water resistance $F_{y}^{\prime }$ on the profile AUV is proportional to the square of the velocity of the profile AUV.

According to Formulas (3) and (5), $f^{\prime }$ is proportional to the current velocity *v_x_*
(9)$$f^{\prime }=8.5\times v_{x}^{2}$$

According to Formulas (1) and (2), $F_{y}^{\prime }$ is proportional to the AUV velocity *v_y_*
(10)$$F_{y}^{\prime }=13.345\times v_{y}^{2}$$

*mg*′ is 4 N after weighing by placing it in the sea water. Due to the slow descent speed of the AUV and the shallow depth of the seawater, the profile AUV descent process is regarded as a uniform acceleration process, and the relationship between the descent time *t* of the profile AUV and the sweater velocity *v_x_* is calculated.
(11)$${mg}^{\prime }-F_{y}^{\prime }-f^{\prime }=ma$$
(12)$$s=\frac{1}{2}at^{2}$$

In Formulas (11) and (12), *a* is the acceleration, *s* is the water depth, and *t* is the profile AUV decent time.

From Formulas (9)–(12), where *m* is 10 kg and *s* is 11 m, the relationship between the seawater flow velocity *v_x_* and the AUV descent time *t* of the AUV can be obtained:(13)$$v_{x}=\sqrt{0.4706-\frac{73.3765}{t^{2}}}$$

According to Formula (13), the flow rate collected by the ADCP is compared with the flow rate calculated according to the profile AUV descent time *t*, as shown in Figure 24.

It can be seen from Figure 24 that when the velocity is greater than 0.3 m/s, the velocity calculated by the AUV descent time is in good agreement with the actual measured velocity, but when the velocity is less than 0.3 m/s, the current velocity cannot be calculated using the AUV descent time. This is due to approximating the AUV descent process as a uniform acceleration, but in fact, due to the resistance of water flow, the AUV descent process is a variable acceleration process. Under the condition that the accuracy of the current velocity is not high, the current velocity calculated using the AUV can be linearly stretched so that the velocity calculated from the AUV descent time has good consistency with the actual measured velocity, as shown in Figure 25.

## 5. Conclusions

This paper presents a profile AUV system suitable for offshore marine surveys. Different from traditional AUVs, which have a large degree of freedom in the horizontal direction, the profile AUVs designed in this study only moved in one dimension in the vertical direction to meet the needs of fixed-point profile measurements in marine surveys. Through the overall system design, including the AUV shape design, AUV dynamic design, and AUV control design, the requirements of multifactor profile surveys in marine survey tidal cycles are satisfied. A multitask Linux operating system is adopted, measuring instruments can be added freely, and all the acquisition parameters can be collected simultaneously. By modeling changes in the depth data during the unpowered AUV descent, the flow velocity of the current is derived, which can save an expensive set of instruments for measuring the current velocity in offshore marine surveys. The wireless data return and remote-control functions designed to provide more flexibility for marine surveys were verified in a docking experiment. In an offshore survey experiment, the profile AUV system measurement results are in good agreement with those of a manual profile survey using a traditional survey ship method. However, the measurement time of the profile AUV system is longer, which can allow the measuring instrument with sufficient response time for each point measurement and make the measurement data more accurate and reliable. The depth change data in the ocean survey are modeled and analyzed to obtain the current velocity, which is in good agreement with the current velocity measured using an ADCP. Therefore, the profile AUV system suitable for offshore marine surveys can simultaneously carry out multiparameter offshore marine profile surveys and measure ocean current velocity. Due to their low cost, multiple sets of high-density and synchronous offshore marine surveys can be deployed within a certain range, greatly reducing the cost of offshore marine surveys.

## Figures and Tables

**Figure 1 sensors-23-03722-f001:**
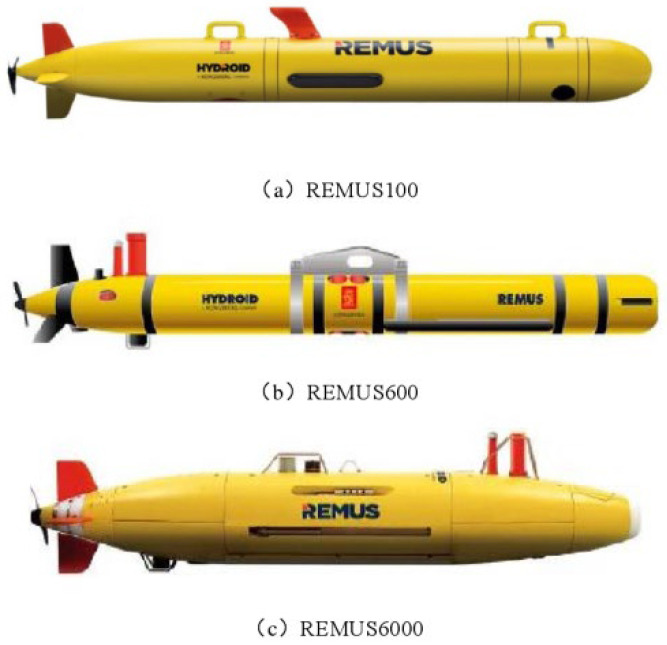
The REMUS series of underwater vehicles.

**Figure 2 sensors-23-03722-f002:**
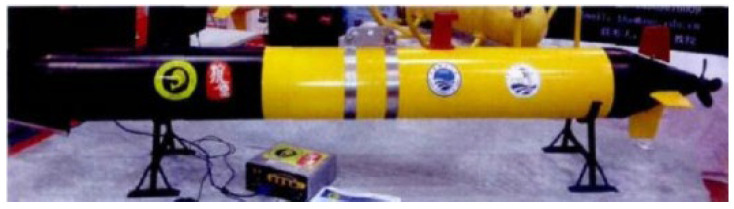
Sailfish underwater vehicle.

**Figure 3 sensors-23-03722-f003:**
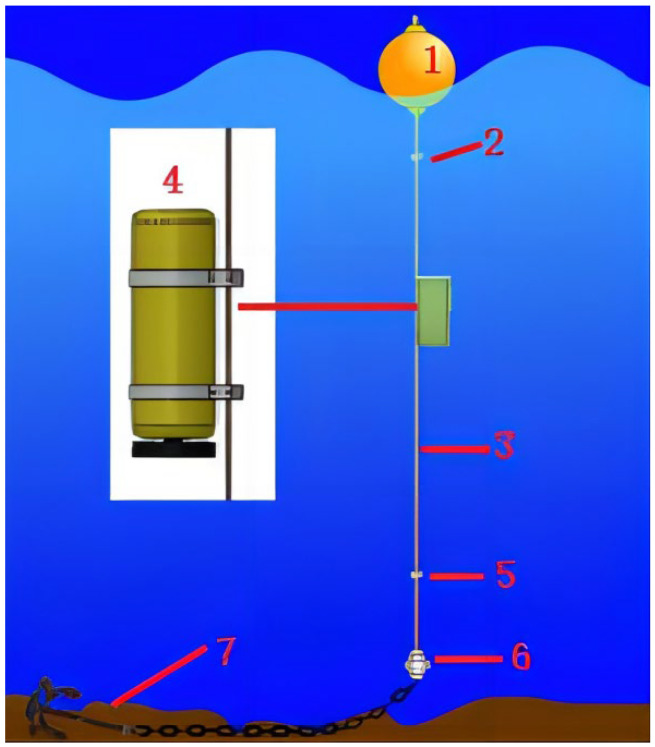
Profile AUV system.

**Figure 4 sensors-23-03722-f004:**
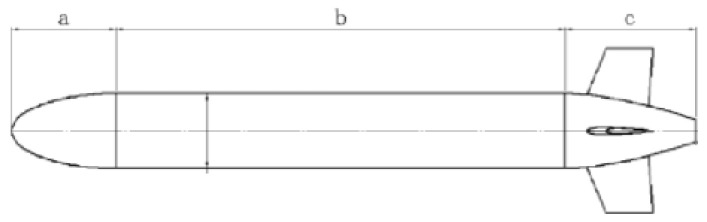
Myring shape outline. a is the forward flow line, b is the middle, c is the stern.

**Figure 5 sensors-23-03722-f005:**
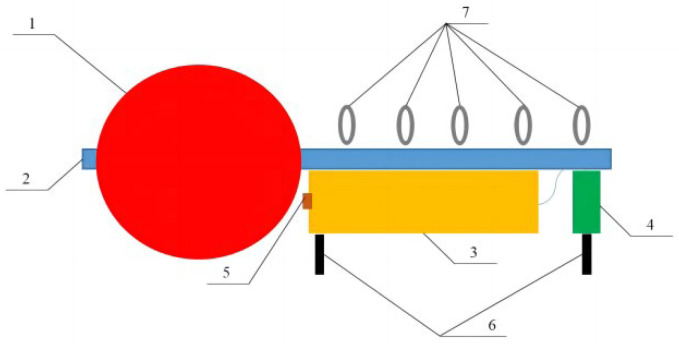
Structure of the profile AUV. 1 is a floating ball, 2 is a fixed column, 3 is middle cabin, 4 is a stern duct propeller, 5 is a temperature-depth sensor, 6 is a sliding connector, 7 is fixed connectors.

**Figure 6 sensors-23-03722-f006:**
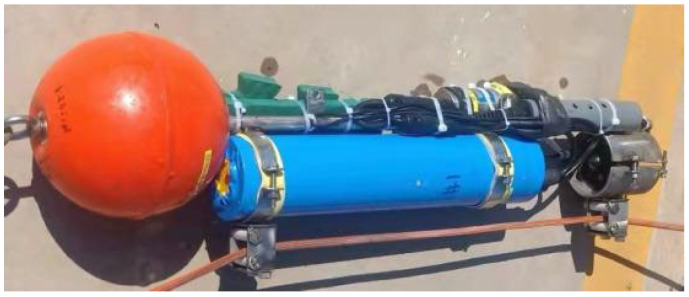
Profile AUV.

**Figure 7 sensors-23-03722-f007:**
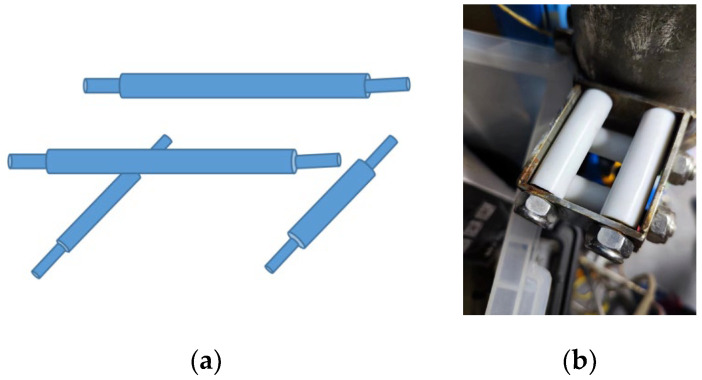
Profile AUV and Kevlar cable connector. (**a**) connector structure drawing. (**b**) connector physical image.

**Figure 8 sensors-23-03722-f008:**
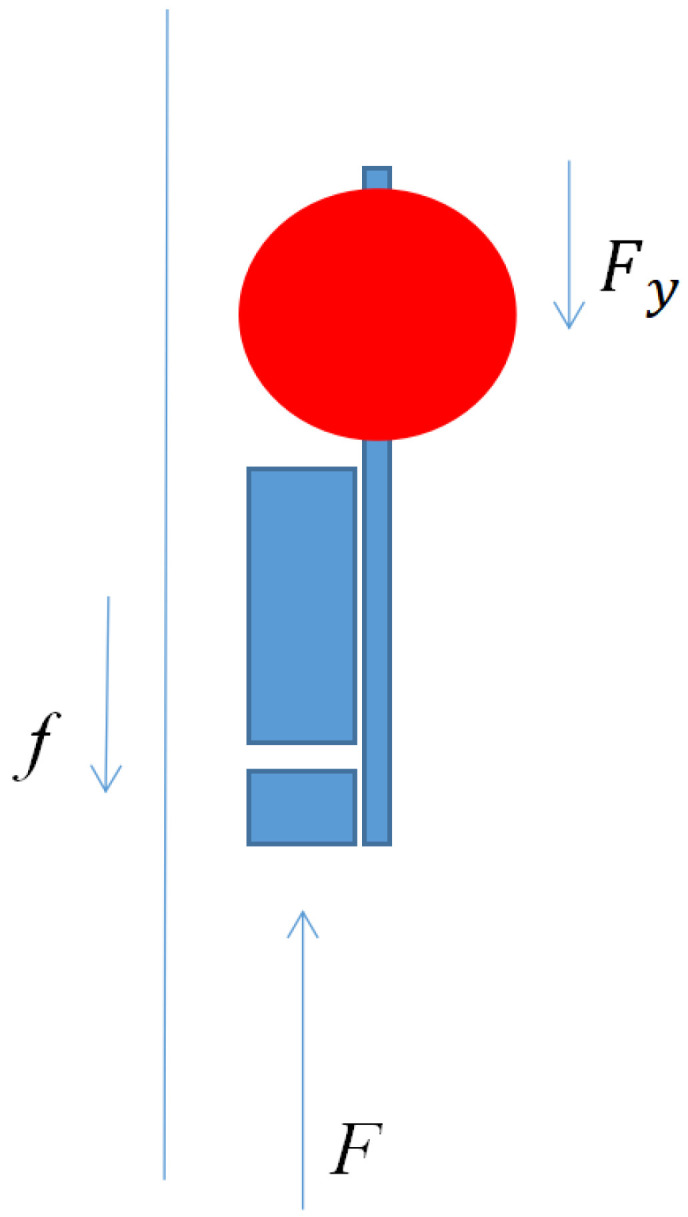
Profile AUV stress in the vertical direction, *F_y_* is the water resistance in the vertical direction, *f* is friction force, *F* is the propeller thrust.

**Figure 9 sensors-23-03722-f009:**
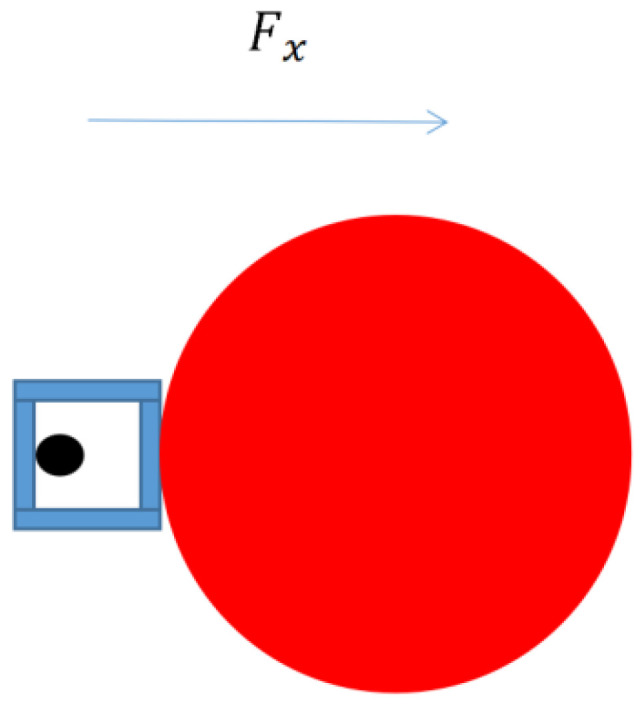
Profile AUV stress in the horizontal direction, *F_x_* is the force of the water flow in the horizontal direction.

**Figure 10 sensors-23-03722-f010:**
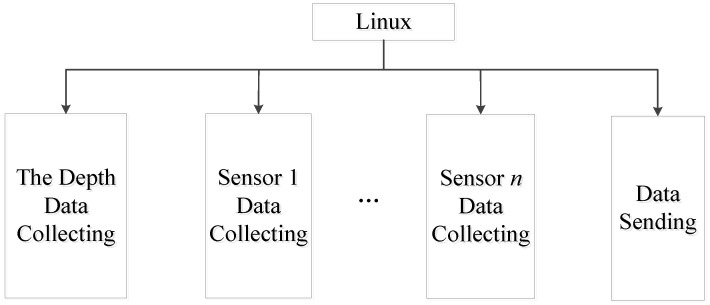
Multitasking controlled using Linux.

**Figure 11 sensors-23-03722-f011:**
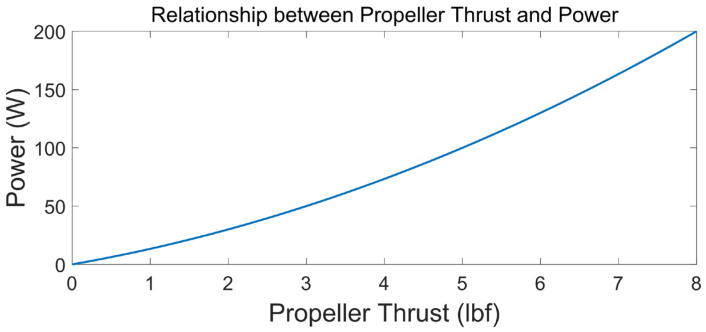
Relationship between the propeller thrust and power.

**Figure 12 sensors-23-03722-f012:**
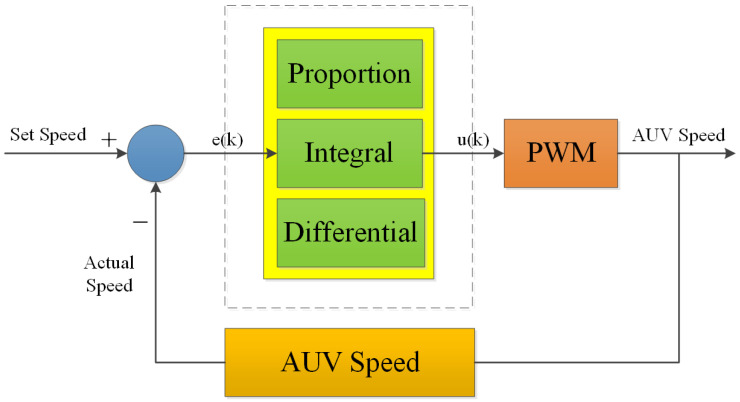
Profile AUV controlled using a PID algorithm.

**Figure 13 sensors-23-03722-f013:**
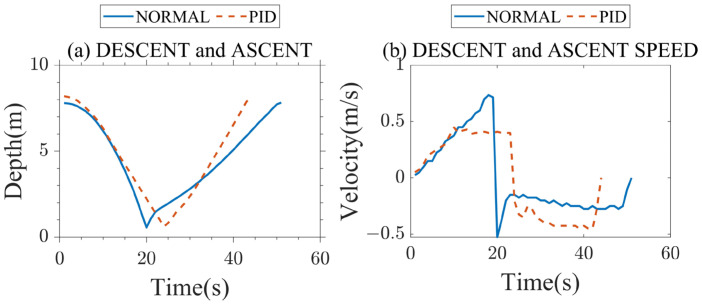
Comparison between NORMAL and PID for profile AUV descent and ascent.

**Figure 14 sensors-23-03722-f014:**
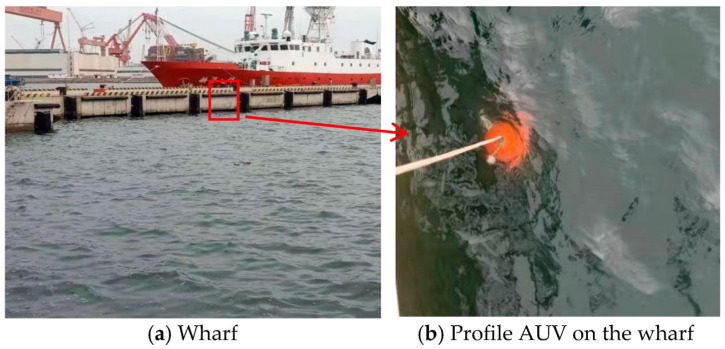
Experiment on the wharf.

**Figure 15 sensors-23-03722-f015:**
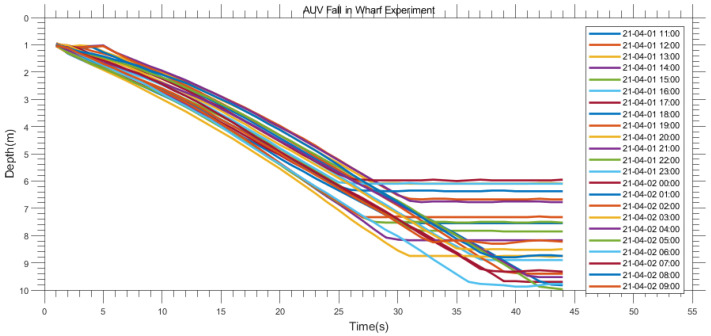
Experimental results at the wharf.

**Figure 16 sensors-23-03722-f016:**
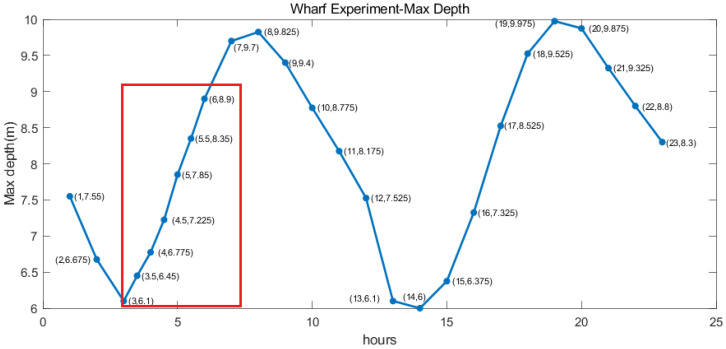
Wharf experiment max depth.the red box part is the acquisition frequency set to twice an hour through remote communication.

**Figure 17 sensors-23-03722-f017:**
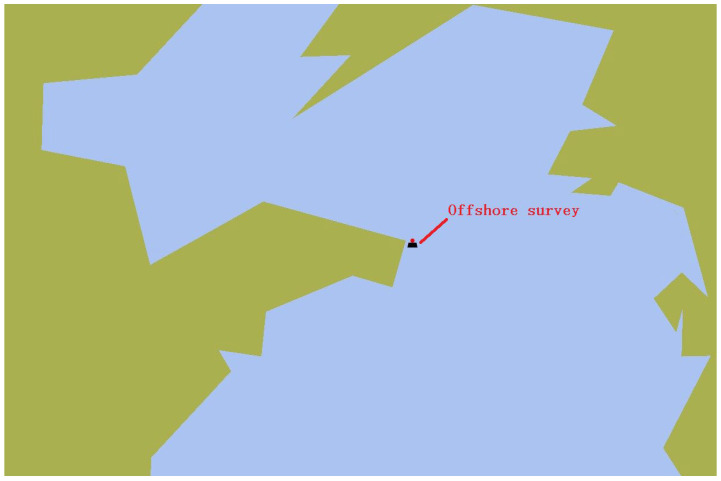
Offshore survey location.

**Figure 18 sensors-23-03722-f018:**
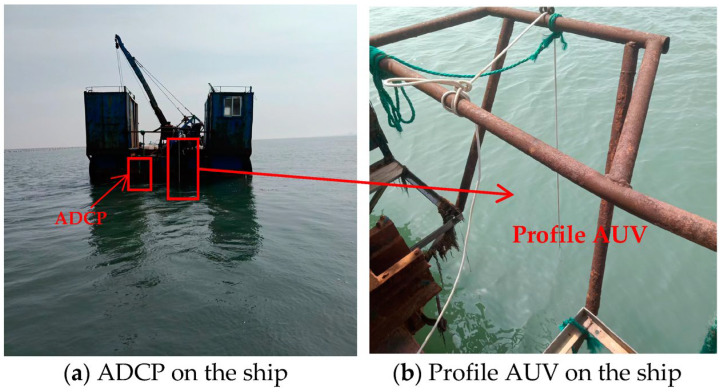
Offshore survey comparative experiment.

**Figure 19 sensors-23-03722-f019:**
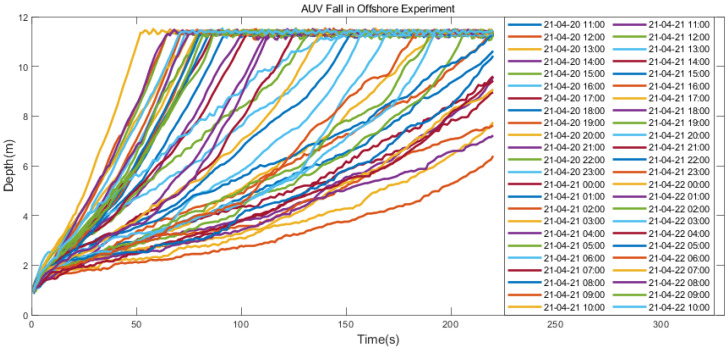
Experimental results of the offshore survey.

**Figure 20 sensors-23-03722-f020:**
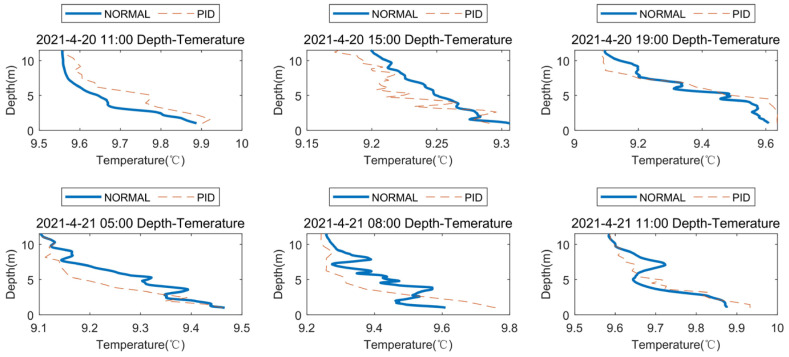
AUV and MANUAL depth-temperature comparison.

**Figure 21 sensors-23-03722-f021:**
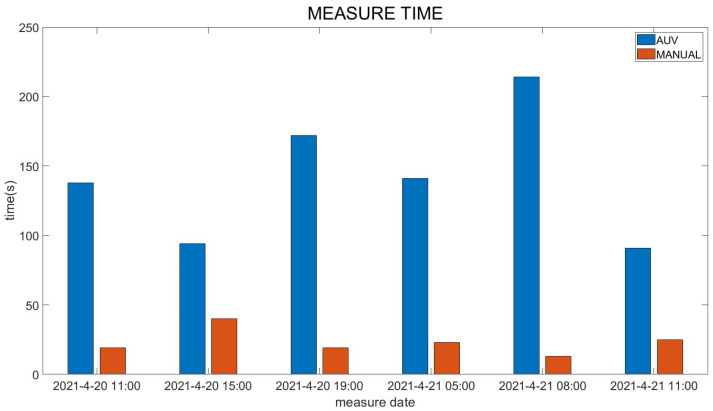
AUV and MANUAL measurement time comparisons.

**Figure 22 sensors-23-03722-f022:**
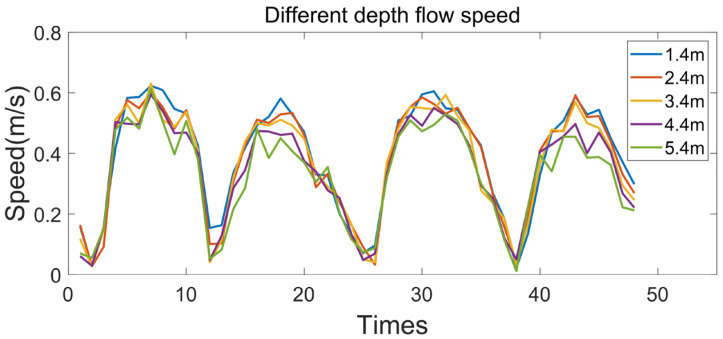
Different depth flow speed.

**Figure 23 sensors-23-03722-f023:**
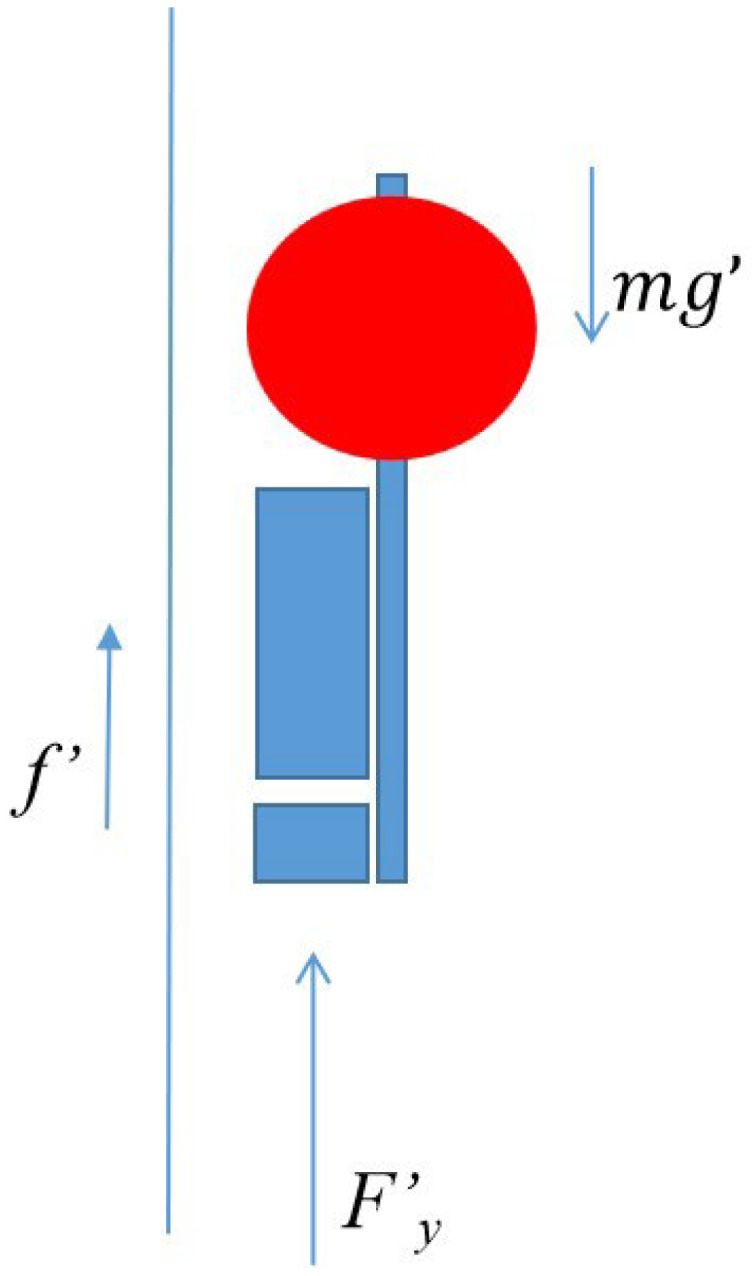
Profile AUV stress in the vertical direction during descent.

**Figure 24 sensors-23-03722-f024:**
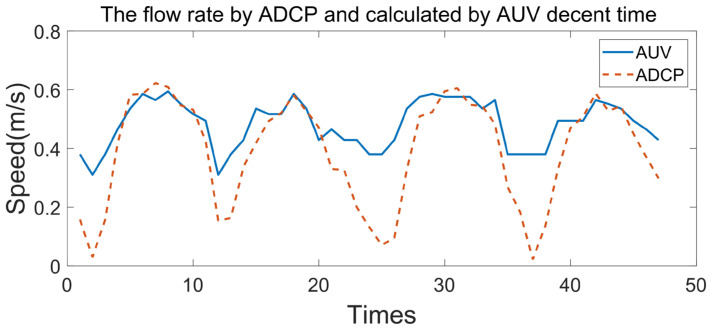
Flow rate using the ADCP and calculated using the AUV descent time.

**Figure 25 sensors-23-03722-f025:**
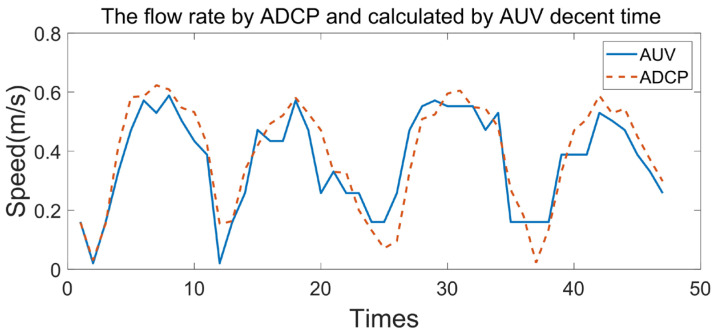
Flow rate using the ADCP and the modified calculation using the AUV descent time.

## Data Availability

The data supporting this study are provided within this paper.

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
