# Peer review of "Profile Autonomous Underwater Vehicle System for Offshore Surveys"

_sensors, 2023, doi:10.3390/s23073722_

Round 1

Reviewer 1 Report

Figure 7 shows that the content is not easy to understand, and it is recommended to add more details and make a distinction.

Why is the place where the data is collected and the place where the experiment is not the same place, and what are the reasons for this?

The innovation points of this paper should be further condensed and highlight the advantages of the designed AUV.

Author Response

Thank you for your review,  please see the attachment file for the response.

Reviewer 2 Report

The paper's english is poor. Major studies in the literature were not examined in detail. You should look at the following work: i) Intelligent-PID with PD Feedforward Trajectory Tracking Control of an Autonomous Underwater Vehicle, KG Zafer Bingul, Machines 11 (2). All equations in the paper described linearly. You should consider nonlinearity in UAV. Control of the UAV uses the  very simple structure of controller. Dynamics of the UAV are not examined and considered. You should state what you have done in the paper. 

Author Response

Thank you for your review. Please see the attachment file for the response.

Reviewer 3 Report

The author presents interesting research on the design and application of a small underwater AUV that can gather data along the vertical line. The solution proposed by the authors seems cheaper and more practical than usual AUV's used for gathering high-density data along the vertical sea column. Therefore I recommend this paper for publication.

List of minor errors and suggestions:

Pg 2., ln 79: instead of writing "During an offshore survey, the current velocity is large," it is better to write "During an offshore survey, the current velocity may be significant,"

Pg 3, ln 113: please reformulate the sentence beginning with "At present, streamlined revolving..." especially the "easy expression of mathematical analysis" part

Pg 4, ln 118: " At present, the main body of an aircraft adopts..." this paper is about AUV, not the aircraft, please reformulate or remove

Figure 8, although f,Fy and F are explained in the text, it would be nice to add these explanations in the figure's legend. The same could be said for Figure 9.

Author Response

(The authors gave the same response as above.)

Round 2

Reviewer 2 Report

The quality and resolution of the Figures in the paper are poor. You should enhanced them. 

Author Response

Please see the attachment. Thank you very much for your comments.
